# Hearing Loss after COVID-19 and Non-COVID-19 Vaccination: A Systematic Review

**DOI:** 10.3390/vaccines11121834

**Published:** 2023-12-09

**Authors:** Xin Wei Liew, Zer Han Malcolm Tang, Yan Qing Cherie Ong, Kay Choong See

**Affiliations:** 1Yong Loo Lin School of Medicine, National University of Singapore, Singapore 169856, Singapore; liewxinwei@gmail.com (X.W.L.); malcolmtangzh@gmail.com (Z.H.M.T.); cherieong379@gmail.com (Y.Q.C.O.); 2Division of Respiratory and Critical Care Medicine, Department of Medicine, National University Hospital, Singapore 119228, Singapore

**Keywords:** vaccine, vaccination, hearing loss, hearing impairment, deaf, deafness

## Abstract

(1) Background: Vaccine safety is an important topic with public health implications on a global scale. The purpose of this study was to systematically review available literature assessing sensorineural hearing loss (SNHL) incidence and severity following both coronavirus disease 2019 (COVID-19) and non-COVID-19 vaccinations, as well as prognosis and outcomes. (2) Methods: This systematic review was performed according to the Preferred Reporting Items for Systematic Review and Meta-Analysis guidelines. Relevant publications evaluating post-vaccination SNHL were selected from PubMed and Embase, searching from inception to July 2023. (3) Results: From 11 observational studies, the incidence of post-vaccination SNHL was low for both COVID-19 and non-COVID-19 vaccines, ranging from 0.6 to 60.77 per 100,000 person-years, comparable to all-cause SNHL. (4) Conclusions: The incidence rates of SNHL following COVID-19 and non-COVID-19 vaccinations remained reassuringly low. Most patients experienced improved hearing function in the weeks to months following vaccination. This study underscores the importance and safety of vaccinations and encourages ongoing surveillance and detailed reporting of hearing loss cases post-vaccination.

## 1. Introduction

Vaccination is one of the best public health interventions in modern times. Vaccines have successfully eradicated debilitating diseases such as smallpox and have also dramatically reduced the incidence rates of other major diseases such as polio and measles [1,2]. Annually, vaccinations are estimated to save 2–3 million lives [1]. With rapidly advancing scientific technologies, almost 30 microorganisms can be targeted with up to 70 vaccines and counting [3].

Since 1796, when Edward Jenner invented the first vaccine against smallpox, vaccinations have saved millions of lives and are indispensable in a physician’s arsenal against microbiological diseases. Most recently, the vaccine has once again been relied upon, specifically for the novel coronavirus, severe acute respiratory syndrome coronavirus-2 (SARS-CoV-2). The World Health Organization (WHO) declared coronavirus disease 2019 (COVID-19) a pandemic on 11 March 2020, and great efforts were invested in producing an effective and safe COVID-19 vaccine [4]. Studies have shown that COVID-19 vaccination has also been pivotal in reducing the morbidity and mortality of COVID-19 patients [5].

Vaccine hesitancy is dangerous and not unique to COVID-19 vaccinations. Concerns about vaccine side effects are the second most common reason driving reluctance to take COVID-19 vaccinations [6,7]. Public concern about vaccine safety is expected and understandable. The WHO identified “reluctance or refusal to vaccinate despite the availability of vaccines” as one of the 10 threats to global health in 2019 [8]. Similarly, measles outbreaks in the United States (where endemic measles is eradicated) are largely contributed by intentional refusal to vaccinate [9]. It is estimated that a 5% decline in measles, mumps, and rubella (MMR) vaccine coverage in the United States would result in an estimated three-fold increase in measles cases for children aged 2 to 11 years annually [10]. This warning sign is found with pertussis, where vaccine hesitancy was linked to an increased risk for pertussis in some populations studied [9].

With the widespread uptake of COVID-19 vaccinations worldwide, otolaryngologic practices saw an increase in the number of anecdotal reports of sensorineural hearing loss (SNHL) post-vaccination [11,12,13]. Specialists in this field encounter increasing challenges in the counseling of such patients who report a temporal association of hearing loss post-COVID-19 vaccination, particularly in terms of the incidence, severity, and prognosis of the hearing loss.

SNHL is defined by the American Academy of Otolaryngology-Head and Neck Surgery as an acute 30 dB hearing loss across three consecutive frequencies as confirmed by audiometry [14], while hearing loss severity is graded based on pure tone audiogram hearing ranges (Table 1) [15]. In this paper, the definition of SNHL is expanded to include 26 dB hearing loss as per Clark et al.’s severity grading and SNHL diagnoses made by clinicians within the individual studies [15]. The annual incidence of SNHL was on average 27 per 100,000 person-years and ranges from 11 to 77 per 100,000 persons per year, depending on age [16]. There are various plausible etiologies for acquiring SNHL, including age-related, noise-related, drug-related, infection/inflammation, trauma, tumors, systemic disorders, vascular disorders, and vaccine-related [17].

The pathogenesis of how the COVID-19 vaccination causes hearing loss is not well understood. Proposed explanations include both the mRNA payload and the lipid nanoparticle delivery vehicle causing auto-immunogenicity [18] as well as the production of immunoglobulin G 10–14 days after vaccine administration, which coincided with SNHL 10–14 days after the vaccination.

A systematic review of the available current literature was therefore conducted to review the incidence and severity of sudden sensorineural hearing loss post-COVID-19 and non-COVID-19 vaccinations. We studied vaccines against hepatitis B, diphtheria, tetanus, measles, mumps, rubella, rabies, and influenza. We aimed to characterize this phenomenon further to guide clinical practice for all physicians, ranging from primary care physicians to otolaryngological clinical practitioners, so that physicians can provide public health messaging to minimize vaccine hesitancy. It is imperative to be up-to-date and transparent about the safety of vaccinations to best promote awareness and ultimately widespread acceptance of vaccines [19].

## 2. Materials and Methods

The study has been registered with PROSPERO (registration number: CRD42023441395). This systematic review was performed according to the Preferred Reporting Items for Systematic Review and Meta-Analysis (PRISMA) guidelines. A comprehensive search of PubMed (pubmed.ncbi.nlm.nih.gov, accessed on 30 July 2023) and Embase (embase.com, accessed on 30 July 2023) was conducted to identify the relevant literature (Figure 1).

The keywords included “vaccine” and “vaccination” AND “hearing loss”, “deaf”, and “deafness”. There was no limit to the timeframe of the search, which was performed on 30 July 2023.

The search produced a list of 561 unique articles. Screening of titles and abstracts was conducted, with analysis of the full texts if there were any doubts as to the suitability of the work for inclusion. We included all observational studies with a description of hearing loss, including those where quantitative audiogram measurements were not listed. A key exclusion criterion included the study population, which already had pre-existing otologic disorders affecting baseline hearing. We have filtered the number of papers to 11.

A qualitative review of the included studies was then performed to uncover a general understanding of the associations of vaccine exposure with hearing loss, as well as the incidence and severity. In addition, interventions to manage hearing loss after vaccination as well as patients’ outcomes were studied.

In our carefully selected observational studies, we reviewed the incidence or prevalence of hearing loss in patients receiving COVID-19 and non-COVID-19 vaccinations. Additional data fields extracted from the full-text documents included the following: patient demographic, vaccine type, number of patients who received the vaccine, time of onset of SNHL since vaccination, associated symptoms, and treatment initiated. XWL, YQCO, and ZHMT did the full text screen and data extraction. XWL and ZHMT assessed observational studies for bias using ROBINS-I (Table 2). Any discrepancies were solved through discussion with the senior author, KCS.

## 3. Results

Out of 444 studies extracted from PubMed and 277 from Embase, we have included 11 observational studies. The PRISMA flowchart is displayed in Figure 1, and the quality evaluation results are displayed in Table 2.

From Table 2, bias across observational studies is mostly “moderate” overall. This is largely due to reporting bias from electronic records, with minimal to no effort in reducing confounders in analysis. Larger observational studies utilize self-reporting systems, as seen with the Vaccine Adverse Event Reporting System (VAERS) in the United States [20,24,25,26] and national healthcare registries in Finland [28], Israel [27], and France [29]. Therefore, reporting bias remains a problem in interpreting the results. Furthermore, confounders play a large role in data analysis, especially when various other demographic and medical factors have to have a direct impact on SNHL, such as cardiovascular risk factors [30]. While the bias has been evaluated as “moderate”, we continued to include these studies due to the fulfillment of our inclusion criteria after filtering from database searches, and we believe they contribute to the available body of evidence pertaining to the limited study of post-vaccination SNHL.

### 3.1. Observational Studies of COVID-19 Vaccines

A total of nine observational studies focused on COVID-19 vaccines (Table 3). The COVID-19 vaccines available use mRNA (e.g., manufactured by Pfizer or Moderna), viral vector-based (e.g., manufactured by Johnson and Johnson), or inactivated vaccine platforms (e.g., manufactured by Sinovac or AstraZeneca). A majority of studies evaluated mRNA-based vaccines only [11,27,29], while others also included viral vector-based vaccines [24,25,26] and inactivated virus vaccines [22,28] in their studies.

Generally, all nine studies showed that the incidence and prevalence of SNHL associated with COVID-19 vaccination were very low, even across different demographics and vaccine types. A range of incidence in large-scale studies can be appreciated, from as low as 0.6 to 28.0 cases per 100,000 person-years. For Yanir et al., however, a study carried out on the Israeli population suggested an increasing trend of SNHL post-vaccination as compared to previous years prior to the vaccination [27]. The paper found an increasing incidence rate (IR) of 60.77 (95% CI, 48.29–73.26) per 100,000 person-years post-COVID-19 vaccination as compared to previous reference years prior to COVID-19 vaccination, which demonstrated an IR of 41.50 (95% CI, 37.98–45.01) per 100,000 person-years in 2018 and 44.46 (95% CI, 40.85–48.07) per 100,000 person-years in 2019. 

The range of prevalence appears to be wide, ranging from as low as 0.00324% in Yanir et al.’s study to 3.85% in Wichova et al.’s study [11,27]. While smaller-scale observational studies conducted by Wichova, Filippatos, and Avci [11,22,23] demonstrated a higher prevalence of SNHL, beginning at 0.2% in Filippatos et al.’s study [23], the larger observational studies from the USA, Finland, France, and Israel show that the nationwide prevalence of SNHL is reassuringly low, with the highest prevalence being 0.0142% in Nieminen’s study [28].

Most studies did not specify the severity of SNHL or discuss more about the recovery and prognosis of those who did suffer from SNHL. Uniquely, hearing loss after COVID-19 vaccination was seen in 1.2% of patients with COVID-19 infection in the past 6 months, as compared to only 0.1% of patients without COVID-19 infection [22].

### 3.2. Observational Studies on Non-COVID-19 Vaccines

Only two large-scale observational studies studied non-COVID-19 vaccines and SNHL. Asatryan et al. studied the measles, mumps, and rubella (MMR) vaccine, and Baxter et al. studied the influenza, tetanus, reduced diphtheria, reduced acellular pertussis, and zoster vaccines [20,21].

In Asatryan et al., the incidence of hearing loss reported after vaccination (1 per 6–8 million doses) appears to be substantially rarer than that seen after natural measles or mumps infection (1 per 20,000 infections) [20]. For Baxter et al., across 7 years of follow-up and over 23 million vaccines, patients with the development of SNHL were not associated with immunization [21]. The severity of SNHL was not described in these two studies.

## 4. Discussion

### 4.1. Overview of Results

This comprehensive systematic review of COVID-19 and non-COVID-19 vaccinations and SNHL aims to better elucidate the complications of administering such vaccines. With ongoing controversy about the effectiveness and safety of vaccines, especially amongst population groups advocating against vaccinations, it is the duty of the medical and scientific community to keep everyone informed on the most accurate and up-to-date data on vaccine safety.

In this systematic review, we found that both the incidence and prevalence of SNHL after COVID-19 vaccinations were low, corresponding to a low disease burden and pressure. The incidence range of SNHL is low, ranging from 0.6 to 60.77 per 100,000 person-years across various cohort studies, and very few case reports on SNHL exist relative to the large number of vaccines administered. In large-scale observational studies, the incidence from all papers reviewed demonstrated that the incidence of SNHL was mostly compatible with the average annual incidence of 27 per 100,000 person-years for all causes, as reported by Alexander et al. in 2013, studying 60 million patients from the United States (US) across 2006–2007 [16]. In nationwide studies conducted in the US, Formeister et al. reported an incidence ranging from 0.6 to 28.0 cases per 100,000 person-years, whereas Chen reported an incidence of 6.66 per 100,000 person-years [24,25]. Similarly, the Finnish study by Nieminen et al. shows an incidence of 21.2 to 22.1 cases per 100,000 person-years, depending on which vaccine was used [28]. Thai-Van et al.’s study similarly demonstrated a small incidence of 1.45 or 1.67 reports per 1,000,000 vaccinations [29]. These large-scale observational studies are in keeping with the average annual incidence of 27 per 100,000 person-years from all causes.

However, there exists an exception in Yanir’s study on the Israeli population in 2022 where the incidence ratio (IR) of SNHL after COVID-19 vaccines was a high of 60.77 (95% CI, 48.29–73.26) per 100,000 person-years, averaged across age groups, and this was comparatively higher than the other observational studies [27]. Across age, the IR increased from 22.44 to 150.53 per 100,000 person-years from age groups 16–44 to patients older than 65. Similarly, Alexander’s study also revealed an increasing incidence of SNHL with age, from 11 per 100,000 for patients younger than 18 years to 77 per 100,000 for patients 65 years and older, and established a positive correlation between age and the incidence of sensorineural hearing loss, which is in keeping with Yanir’s study [16,27]. Additionally, Yanir’s relatively high IR of SNHL is also found in previous reference years prior to the COVID-19 pandemic in the same paper: 41.50 (95% CI, 37.98–45.01) per 100,000 person-years in 2018 and 44.46 (95% CI, 40.85–48.07) per 100,000 person-years in 2019, suggesting that the baseline incidence of SNHL in the Israeli population is already above the average proposed by Alexander [16,27]. Yanir noted in his study that people who were vaccinated were older and may be sicker than the reference population, with a mean age of 46.8 ± 19.6 years [27]. Many reasons account for this higher IR of post-vaccination SNHL in Yanir’s cohort relative to similar studies [27]. These include the inherent differences between the Israel population and other study populations and many health confounders that were not accounted for. In particular, Yanir noted that cardiovascular risk factors as well as coagulation disorders, which are themselves risk factors for SNHL, were not accounted for in the study and stated that the lack of data detailing the health characteristics of the exposure group was a serious limitation [27]. Additionally, Alexander’s incidence, used as a reference in this paper, was calculated with patients from 2006–2007, and changes in health-seeking behavior over time could be attributed to the stark difference [16]. Fortunately, Yanir’s study concludes with a small attributable risk (AR) to post-vaccination SNHL, with the highest AR of 3.74 per 100,000 vaccinees, and concludes that the influence on public health would be relatively minor [27].

In smaller observational studies ranging from 500 to 1710 participants, the incidence of SNHL was not reported, and there was no period of follow-up in those studies. Hence, these studies were not taken into account when reviewing the incidence.

All observational studies on COVID-19 vaccines showed that the incidence of SNHL associated with COVID-19 vaccination is reassuringly low, even across different demographics and vaccine types (Table 3). Similarly, the two studies on non-COVID-19 large-scale vaccination campaigns, such as the live attenuated MMR vaccine, the inactivated influenza vaccine, tetanus, reduced diphtheria, and reduced acellular pertussis (Tdap) vaccine, as well as the zoster vaccine, also did not demonstrate an increased incidence of SNHL in the general population (Table 4) [20,21].

The prevalence of SNHL after the COVID-19 vaccination appears reassuringly low as well. For the large-scale observational studies, the three studies comparing SNHL to the number of participants affected reflect a small range of prevalence from 0.00350% to 0.0142%. Nieminen’s Finnish study reflected the highest prevalence, where 0.0142% of participants were found to have SNHL after the Pfizer-BioNTech COVID-19 vaccination [28]. Chen’s study follows next, with a prevalence of 0.00666% [24]. Lastly, Yanir’s Israeli study had the lowest prevalence of 0.00350% after the first dose of the COVID-19 vaccination [27].

For the other two large-scale observational studies comparing SNHL to the number of doses, the prevalence of SNHL was low as well. In Formeister’s study, the prevalence was 0.16 cases per 100,000 doses for both the Pfizer-BioNTech and Moderna vaccines and 0.22 cases per 100,000 doses for the Janssen/Johnson and Johnson vaccine [25]. In Thai-Van’s French study, the prevalence ranges from 0.000128% to 0.000145%, depending on the vaccine used [29].

In Guo’s study comparing the adverse effects of the COVID-19 vaccination, the prevalence of deafness and hypoacusis only accounted for a small percentage, with 809 incident reports of deafness and 781 reports of hypoacusis out of 717,577 reported vaccination adverse effects [26]. Both account for only 0.1% of all vaccination adverse effects. Hence, even the prevalence of hearing loss within the pool of reported adverse effects post-COVID-19 vaccination is extremely low.

The prevalence of SNHL after COVID-19 vaccination in smaller observational studies appears to be significantly higher. In Filippatos’ study of 502 healthcare workers, there was one case of SNHL, leading to a prevalence of 0.2% [23]. In Avci’s study of 1710 healthcare workers, there were five reports of SNHL, giving a prevalence of 0.3%. Lastly, in Wichova’s study, 40 of the 1641 patients and 51 of the 1325 patients who visited the clinic, respectively, in 2020 and 2021, were found to have clinically diagnosed SNHL, giving a prevalence of 2.44% (in 2020) and 3.85% (in 2021) [11,22]. While these prevalences appear to be higher than those reported in the large-scale observational studies, this can be largely accounted for due to a small sample size, which does not accurately represent the entire population, as well as selection bias. Avci’s and Filippatos’ studies were carried out exclusively on healthcare workers, which is epidemiologically not representative of the general population [22,23]. For Wichova’s study, it too follows that there would be a proportionally greater number of individuals presenting at otolaryngologic clinics or participating in interviews who have hearing loss as compared to the general population, creating a selection bias [11]. The results may also be confounded by predisposing otolaryngologic pathologies, which may explain the hearing loss.

From the above, there does not appear to be a correlation between SNHL and COVID-19 vaccination, with the incidence as well as prevalence of SNHL post-vaccination being low across many large observational studies on different geographical populations. As such, the burden of SNHL post-vaccination, if any, is low and will likely remain low in time to come.

While incidence remained low across time and between countries, an observational study by Wichova et al. noted a pattern of increase in incidence since the pandemic [11]. Following the pandemic onset in early 2020 to the present, there has been a clear increase in this diagnosis, with a more than two-fold increase to 2.44 and 3.85% in 2020 and 2021, respectively. While an increased incidence does not by itself prove causation, the trend here does bring up concern that in some patients, there may be a post-vaccination change in hearing. One study compared the incidence of SNHL between the different vaccines [24]. Chen et al. identified increased risk for hearing disorder following administration of COVID-19 vaccines (both mRNA and virus vector) compared to influenza vaccination in real-world settings [24]. While incidences within each vaccine remain low and insignificant, the inter-vaccine differences could hold immunologic and biological mechanisms to uncover. 

Our systematic review also explored the frequency and distribution of age in SNHL post-vaccination by comparing the mean age as well as the age range of the study populations at the time of vaccination. One intriguing aspect of this discussion is the trend observed in some studies, suggesting a rise in the age of individuals experiencing SNHL post-COVID-19 vaccination. Publications reported that the mean age at the time of vaccination is most prevalent in individuals greater than 45 years old [11,23,25,27,29,31,32]. This is consistent with the health patterns of the general population, in which SNHL prevalence increases with age. It is essential to consider age-related hearing loss as a baseline, as older adults may experience hearing loss coincidentally with receiving the vaccine, making it challenging to establish a direct causation. This trend raises critical questions about the interplay between age, vaccination, and hearing health.

There are various case reports and series on patients suffering from SNHL post-COVID-19 vaccination. Most reported mild to moderate hearing loss with complete recovery after corticosteroids, but there remained a handful reporting more severe SNHL, which had only a partial response to treatment. However, these case reports and series were excluded from our analysis, as these publications are primarily descriptive with no long-term follow-up data. The level of evidence is low, and it is likely that these reports represent a biased subgroup, where there is a risk of reporting bias. Our comprehensive study of observational studies, on the other hand, offers a broader perspective by analyzing patterns and trends across a larger population. This observational data provides a statistical basis to draw conclusions about the prevalence and incidence of SNHL post-vaccinations spanning across diverse demographics and may give a better perspective on the issue of SNHL in light of these case reports. Nonetheless, case reports are invaluable for elucidating rare conditions and atypical presentations of hearing deficits post-vaccinations. Kahn et al. reported on an alarming case of a young, 20-year-old male with bilateral profound SNHL as part of a multisystem inflammation and organ dysfunction of unknown mechanism after administration of the Pfizer COVID-19 vaccine [31]. In this patient, acute stroke, pericardial effusion and tamponade, pleural effusion, and acute kidney injury were described. This detailed narrative provides a foundation for further investigation and sheds light on the devastating, albeit rare, complications that can arise post-vaccination. Regardless, case reports suggest that the prognosis for post-vaccination SNHL was generally favorable, with frequent reversibility and partial to complete recovery in most cases.

### 4.2. Limitations of Our Study and Literature

Our systematic review has several limitations that should be acknowledged. For each vaccine, the number of studies available is limited, with few observational studies. Nevertheless, there is internal consistency in the overall conclusion of the papers included, and no prospective studies or randomized controlled trials exist for inclusion to dissuade our conclusion.

Another limitation is the inadequate data presented in the included studies. On top of the inherent bias of such articles evaluated as “moderate”, these large-scale studies do not describe the severity, duration, and prognosis of SNHL post-vaccination. Importantly, the time of onset from SNHL. Additionally, another limitation is that, as the above studies are retrospective in nature, we cannot definitively conclude the causation between the vaccination and SNHL. Hence, we would recommend large-scale randomized control trials to support our theory and establish a more concrete understanding of the adverse effects of vaccines. Future studies and trials of vaccine safety should specifically check for this complication through objective hearing screening, with confirmation and data registration with pure tone audiometry in symptomatic patients.

A special consideration highlighted is the use of COVID-19 vaccines in a patient who previously contracted the COVID-19 virus itself. Avci et al. showed that the incidence of otolaryngology-specific symptoms such as hearing loss may be higher after inactivated COVID-19 vaccination in patients who were already previously infected by COVID-19 [22]. It may be prudent to inform such patients of a potentially increased risk relative to the general population before receiving an inactivated COVID-19 vaccine. The papers studied in our systematic review did not stratify the population into patients with prior COVID-19 infection compared to those without, and this may be a significant confounder for further researchers to elucidate its influence, even by non-COVID-19 infective agents.

Fortunately, there is nothing to suggest a direct association between SNHL and the administering of vaccines themselves, and vaccination campaigns with strong uptake in vaccination in the name of public health should continue to be encouraged. While there could be a potential link between vaccinations and SNHL, the evidence is largely anecdotal, and no correlation has been proven so far. Therefore, the benefit of mass vaccination in the aftermath of the SARS-CoV-2 pandemic remains unchallenged. As physicians responsible for the long-term health of our patients, information and the prognosis of the uncommon SNHL are important for us to aid our patients. While the incidence is fortunately low, future studies and reports on such complications should include detailed data on the illness for us to minimize the debilitating effects that deafness could potentially have. Another nuance to consider is that the available data primarily represents early post-vaccination periods, and the long-term effects of vaccinations (especially novel ones like COVID-19 vaccinations) on hearing loss are yet to be fully understood. Additionally, the heterogeneity of vaccine types, dosages, and booster scheduling across the included reports further complicates the interpretation of the findings.

### 4.3. Future Directions

Vaccine development, approval, and public acceptance are often a long process in which up-to-date and comprehensive data on potential complications and adverse effects is paramount to protecting society from diseases. We are relieved to have found low incidences of post-vaccination SNHL. Subsequently, various suggestions are raised to better elucidate the unknown variables. This can include more detailed logging of the severity of SNHL as well as its recovery and prognosis. More factors (largely under-recorded) with utility include the confounding effect of previously infected patients and various permutations of vaccination status (in terms of the number of booster shots received and duration between vaccinations). We theorize that deeper analysis of such factors can uncover unknown associations with adverse effects of vaccinations to better direct vaccine indications and even scheduling. We look forward to large-scale prospective randomized controlled trials with meaningful stratification of age, sex, and medical co-morbidities to conclude a causative effect more strongly between vaccinations and SNHL.

## 5. Conclusions

In conclusion, this review of 11 observational studies demonstrated a minimal co-relationship between vaccinations and SNHL. The incidence, prevalence, and hence burden of SNHL post-COVID-19 vaccinations remain small across many different nations. The majority of the observational studies report an incidence that falls within the average annual incidence of SNHL of 27 per 100,000 person-years for all causes in a large US study [16]. The prevalence of SNHL remains reassuringly low with the exception of small-scale observational studies, which can be accounted for by sampling bias and selection bias. Hence, the burden on SNHL is small and is likely to remain small with time.

Unilateral hearing loss seems to be more common than bilateral, and alert physicians can rely on the speedy usage of steroids as a safe and reliable treatment for SNHL to likely ameliorate patients’ hearing impairment post-vaccination. Thankfully, the majority of patients will return to their normal level of hearing within weeks or months. Vaccinations and their protection for the global community strongly outweigh the weakly related otologic complications. International collaboration between otolaryngologists, immunologists, and vaccine researchers would further strengthen our knowledge in the area of post-vaccination hearing loss.

## Figures and Tables

**Figure 1 vaccines-11-01834-f001:**
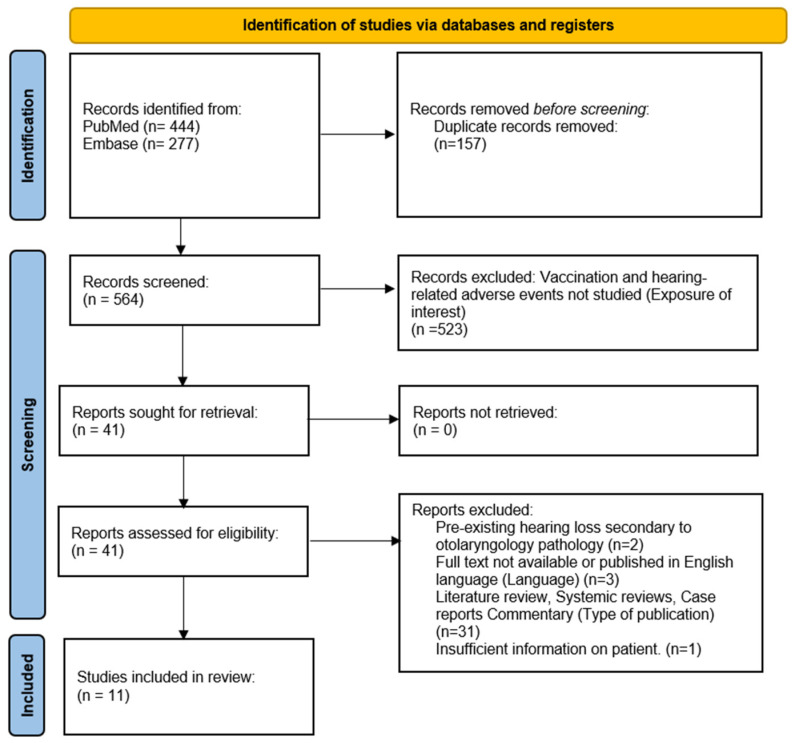
PRISMA flow diagram.

**Table 1 vaccines-11-01834-t001:** Severity of hearing loss based on audiogram metrics.

Hearing Range	dB
Normal	−10–25
Mild	26–40
Moderate	41–55
Moderately Severe	56–70
Severe	70–90
Profound	91+

dB: decibel.

**Table 2 vaccines-11-01834-t002:** Risk of bias assessment of observational studies using ROBINS-I.

Study Author (Year of Publication) (Ref.)	Vaccine Type	Bias Due to Confounding	Bias in Selection of Participants	Bias in Classification	Bias Due to Deviations	Bias Due to Missing	Bias in the Measurement of Outcomes	Bias in Reporting of Data	Overall Risk of Bias
Astrayan (2008) [20]	MMR	Moderate	Moderate	Low	NI	Moderate	Moderate	Low	Moderate
Baxter (2016) [21]	Influenza, tetanus, reduced diphtheria, reduced acellular pertussis, and zoster	Moderate	Low	Low	NI	Moderate	Moderate	Low	Moderate
Avci (2021) [22]	COVID-19	Moderate	Low	Low	Low	Low	Low	Low	Low
Filippatos 2021 [23]	COVID-19	Moderate	Low	Low	NI	Moderate	Moderate	Low	Moderate
Wichova (2021) [11]	COVID-19	Moderate	Low	Low	Low	Moderate	Low	Low	Low
Chen (2022) [24]	COVID-19	Moderate	Low	Low	Low	Low	Low	Low	Low
Formeister (2022) [25]	COVID-19	Moderate	Low	Low	NI	Moderate	Moderate	Low	Moderate
Guo (2022) [26]	COVID-19	Moderate	Low	Low	NI	Moderate	Low	Low	Moderate
Yanir (2022) [27]	COVID-19	Moderate	Low	Low	NI	Moderate	Low	Low	Moderate
Nieminen (2023) [28]	COVID-19	Moderate	Low	Low	NI	Moderate	Low	Low	Moderate
Thai-Van (2023) [29]	COVID-19	Moderate	Low	Low	NI	Moderate	Low	Low	Moderate

NI = no information.

**Table 3 vaccines-11-01834-t003:** Observational studies of SNHL in patients after taking COVID-19 vaccines.

Vaccine Type	Study Author (Year of Publication)	Number of Participants/Doses	Mean Age at the Time of Vaccination	Incidence of SNHL	Prevalence of SNHL	Severity of SNHL	Prognosis
COVID-19 (mRNA and viral vector) (Pfizer-BioNTech, Moderna, or Janssen/Johnson and Johnson)	Chen (2022) [24]	224,660,453 participants from the Vaccine Adverse Event Reporting System (VAERS). Hearing impairment is defined as SNHL, aural fullness, and tinnitus.	NI	6.66 per 100,000 person-years (14,956 reports over a ~2-year period of follow-up)	0.00666% (14,956 reports)	NI	NI
COVID-19 (mRNA and viral vector) (Pfizer-BioNTech, Moderna, or Janssen/Johnson and Johnson)	Formeister (2022) [25]	185,424,899 doses from the Vaccine Adverse Event Reporting System (VAERS)	54 years	0.6 to 28.0 per 100,000 person-years (555 reports over a 7-month follow-up period)	0.16 cases per 100,000 doses of both the Pfizer-BioNTech and Moderna vaccines 0.22 cases per 100,000 doses of Janssen/Johnson and Johnson vaccines	NI	8 of 14 patients with posttreatment audiometric data experienced improvement after receiving treatment
COVID-19 (mRNA) (Pfizer-BioNTech)	Yanir (2022) [27]	Israel First dose 2,602,557 participants	46 years	60.77 per 100,000 person-years (91 reports over a 6 month follow-up period)	0.00350% (91 reports)	NI	NI
		Second dose 2,441,719 participants		56.24 per 100,000 person-years (79 reports over a 6 month follow-up period)	0.00324% (79 reports)		
COVID-19 (mRNA) (Pfizer-BioNTech of Moderna)	Thai-Van (2023) [29]	France 97,840,529 doses of Pfizer 22,690,889 doses of Moderna from the Natural Healthcare Registry	51 years (for Pfizer) 47 years (for Moderna)	Pfizer	Pfizer	NI	NI
				1.45 per 1,000,000 injections	0.000145% (142 cases in 97,840,529 doses)		
				Moderna	Moderna		
				1.67 per 1,000,000 injections	0.000128% (29 cases in 22,690,889 doses)		
COVID-19 (mRNA and inactivated) (Pfizer-BioNTech, Moderna, or AztraZeneca)	Nieminen (2023) [28]	Finland ~5,500,000 individuals from the respective national registry	NI	AstraZeneca	AstraZeneca	NI	NI
				22.1 per 100,000 person-years (71 reports over ~2-year period of follow-up)	0.00130% (71 reports)		
				Pfizer	Pfizer		
				21.2 per 100,000 person-years (779 reports over ~2-year period of follow-up)	0.0142% (779 reports)		
				Moderna	Moderna		
				18.5 per 100,000 (188 reports over ~2-year period of follow-up person-years	0.00452% (188 reports)		
COVID-19 (inactivated) (CoronaVac or Sinovac Life Sciences)	Avci (2021) [22]	1710 participants (healthcare workers)	35.79 years	NI	0.3% (5 reports)	NI	NI
COVID-19 (mRNA) (Pfizer)	Filippatos (2021) [23]	502 participants	48.17 years	NI	0.2% (1 report)	NI	Improvement noted
COVID-19 (mRNA) (Pfizer-BioNTech or Moderna)	Wichova (2021) [11]	1641 clinical visits (in 2020)	60.9 years	NI	2020 2.44% (40 reports)	The mean pure tone average (PTA) was 52.2 ± 30.6 dB HL for the affected ear and 21.2 ± 12.5 dB HL for the unaffected ear. The word recognition score (WRS) was 60.6 ± 38% for the affected ear and 90 ± 23.0% for the unaffected ear.	NI
		1325 clinical visits (in 2021)			2021 3.85% (51 reports)		
COVID-19 (mRNA and viral vector) (Pfizer-BioNTech, Moderna, or Janssen/Johnson and Johnson)	Guo (2022) [26]	717,577 adverse effects from the Vaccine Adverse Event Reporting System (VAERS)	NI	NI Of the 717,577 vaccine adverse effects: Deafness: 809 incident reports, PRR 2.03 Hypoacusis: 781 incident reports, PRR 2.50	NI	NI	NI

NI = no information.

**Table 4 vaccines-11-01834-t004:** Observational studies of SNHL in patients after taking non-COVID-19 vaccines.

Vaccine Type	Study Author (Year of Publication)	Number of Participants/Doses	Mean Age at the Time of Vaccination	Description of Incidence, Incidence Ratio (IR), Odds Ratio (OR) of SNHL	Prevalence Rate Ratio (PRR) of SNHL	Severity of SNHL	Prognosis
MMR	Asatryan (2008) [20]	168–224 million doses of MMR vaccine from 1990 to 2003	16 months	NI	59 cases—14 from VAERS and 15 from case reports; C1 case per 6–8 million doses	NI	NI
Influenza, tetanus, reduced diphtheria, reduced acellular pertussis, and zoster	Baxter (2016) [21]	8,354,237 doses given from 2007 to 2013	NI	ORs for vaccination 1 week prior to SSHL were: 0.965 (95% CI, 0.61–1.50) for trivalent inactivated influenza vaccine (TIV), 0.842 (95% CI, 0.39–1.62) for tetanus, reduced diphtheria, and reduced acellular pertussis, and 0.454 (95% CI, 0.08–1.53) for the zoster vaccine.	NI	NI	NI

NI = no information.

## Data Availability

Data supporting this review are available in the reference section.

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
