# Peer review of "Hearing Loss after COVID-19 and Non-COVID-19 Vaccination: A Systematic Review"

_vaccines, 2023, doi:10.3390/vaccines11121834_

Round 1

Reviewer 1 Report

Comments and Suggestions for Authors

The study is a systematic review assessing hearing loss incidence and severity for 34 Covid-19 and non-Covid-19 vaccination studies. Studies focused on Covid-19 vaccines. Among non-Covid 19 vaccination studies, authors studied the effects of vaccines against hepatitis B, diphtheria, tetanus, measles, mumps, rubella, rabies, and influenza.

Some points require clarifying:

Tables 2 and 3: please add whether there are Covid or non-Covid vaccinations.

Table 3: please add explanatory legend linking the table with the questions.

Table 5 and related results: how long after the vaccine, was hearing loss observed? (to keep in line with the characteristics reported in Covid-vaccine studies).

Figure 2: the explanation of the numbers in the graph is required (mention that it is the n number for that condition). The design of that figure requires improving, in particular the lines curvatures require changing to straight lines.

The authors acknowledge the complication of the interpretation and that the study has got limitations such as the limited number of studies available and the small number of patients in case reports. Their study, however, provides insight on the safety of vaccinations and is therefore of scientific interest.

Author Response

Thank you Sir/Madam for your review and comments.

Point 1: Added as per suggested

Point 2: case reports have been removed after internal discussion and review.

Point 3: reviewed that onset of hearing loss for observational studies were insufficiently presented. Has elaborated on this aspect as limitation of available evidence.

Point 4: Figure 2 has been removed after internal discussion and review.

Reviewer 2 Report

Comments and Suggestions for Authors

I recommend major revision of the manuscript “Hearing loss after COVID-19 and non-COVID-19 vaccination: A systematic review”.

Overall, while I found the topic to be important and of interest, the paper seems fragmented and the important questions sought to be addressed are lost.

My reasons for this are as follows –

1.       Inadequate quality assessment (Prisma 2020 Statement 14, 21,22).  Beyond Table 2, there is little real discussion of the nature or impact of study bias despite indicating that confounding was moderate in all studies and the overall risk of bias was judged to be moderate in the majority of studies.  The topic of bias is mentioned at the end of the discussion where a comparison is made of control of bias in the smaller cohort studies compared to the larger population-based studies VAERS.  I would recommend expanding and moving this into the body of the systematic review.

2.       Inadequate results including a failure of syntheses (Prisma 2020 Statement 20a, 20c, 20c, 21, 22).

·         The paper compares rates and metrics across studies with vastly different numbers of participants and designs (population registries, cohort studies of healthcare workers).  These are very heterogenous studies!  I appreciate there may be low rates of hearing loss after vaccination but this analysis doesn’t provide convincing evidence that’s the case.  A discussion of rates is taken up somewhat in the discussion section on page 17 but the point regarding the evidence for low rates of SNHL. hearing loss is lost in the detail.  I would recommend an extracted table summarizing the rates for the relevant study.

·         The role of age in the rates of SINL is lost – the ages of healthcare workers will differ considerably from the VAERS reporting.

·         The discussion mentions two large population-based studies that have been recently reported.  It seems that their incorporation with the exclusion of smaller studies of healthcare workers (Filippatos or Avci) would be an improvement and could be methodologically justified. 

3.       Case reports – I think the paper would be stronger without the case report section which I don’t find add a lot to the important question of whether vaccination results in hearing loss.  Most of the manuscript space is devoted to simple descriptions of the studies without any discussion of their quality (Table 3) and little of their results (lines 359-269).  Figure 2 is not helpful.   

Comments on the Quality of English Language

The manuscript is written in clear English.  

Author Response

Thank you Sir/Madam for your review and comments. We have made an effort to revise our paper accordingly and we look forward to future refinement.

Point on inadequacy of results and failure of synthesis.

  • Tables have been refined to highlight prevalence and incidence for a cohesive and concise presentation amidst the heterogeneity.
  • Discussion segment has been expanded to explain incidence and prevalence in a more focused manner.

Point of age of SNHL

  • Tables have been expanded to include ages of participants.
  • Discussion segment has been expanded to elaborate on the point on age as well.

Point on inclusion of recent reports and exclusion of smaller studies.

  • Edited as recommended
  • All case reports and Figure 3 are excluded.
  • Discussion segment briefly shared on existing case reports without having it as a focus of our study.

Reviewer 3 Report

Comments and Suggestions for Authors

The authors wrote a review about Hearing loss after COVID-19 and non-COVID-19 vaccination. The review should be interesting, but I think that the analysis of the result is poor, without a statistical correct analysis.

In the Prisma floor chart, you arrived at 38 manuscript, exluded 7. They should be 31 and not 34.

The flow chart of reports (figure 2) is very confused, please summeriz it, maybe in a table.

Regarding conclusion, I think that the number of manuscript and cases are very poor, so no serious conclusions can be drawn.

Author Response

Thank you Sir/Madam for your review and comments.

Point 1: Edited accordingly.

Point 2: Figure 2 has been removed after internal discussion and review.

Point 3: Case reports have been removed as a focus of discussion in our paper. 

Round 2

Reviewer 2 Report

Comments and Suggestions for Authors

I am satisfied with the revision. The authors do occasionally refer to this work as a systemic rather than a systematic review.  The tables containing the study summaries remain awkward and difficult to read. 

Author Response

Dear Sir/Ma'am

Thank you so much for your time and review.

We have revised consistency for "systematic review"

The table have been expanded with columns for mean age, incidence and prevalence.